# Efficacy of Low-Cost Sensor Networks at Detecting Fine-Scale Variations in Particulate Matter in Urban Environments

**DOI:** 10.3390/ijerph20031934

**Published:** 2023-01-20

**Authors:** Asrah Heintzelman, Gabriel M. Filippelli, Max J. Moreno-Madriñan, Jeffrey S. Wilson, Lixin Wang, Gregory K. Druschel, Vijay O. Lulla

**Affiliations:** 1Department of Earth Sciences, Indiana University-Purdue University Indianapolis (IUPUI), Indianapolis, IN 46202, USA; 2Environmental Resilience Institute, Indiana University, Bloomington, IN 47408, USA; 3Department of Global Health, DePauw University, Greencastle, IN 46135, USA; 4Department of Geography, Indiana University-Purdue University Indianapolis (IUPUI), Indianapolis, IN 46202, USA; 5Independent Researcher, Indianapolis, IN 46214, USA

**Keywords:** PM_2.5_, citizen science, low-cost sensor, PA, tree canopy coverage

## Abstract

The negative health impacts of air pollution are well documented. Not as well-documented, however, is how particulate matter varies at the hyper-local scale, and the role that proximal sources play in influencing neighborhood-scale patterns. We examined PM_2.5_ variations in one airshed within Indianapolis (Indianapolis, IN, USA) by utilizing data from 25 active PurpleAir (PA) sensors involving citizen scientists who hosted all but one unit (the control), as well as one EPA monitor. PA sensors report live measurements of PM_2.5_ on a crowd sourced map. After calibrating the data utilizing relative humidity and testing it against a mobile air-quality unit and an EPA monitor, we analyzed PM_2.5_ with meteorological data, tree canopy coverage, land use, and various census variables. Greater proximal tree canopy coverage was related to lower PM_2.5_ concentrations, which translates to greater health benefits. A 1% increase in tree canopy at the census tract level, a boundary delineated by the US Census Bureau, results in a ~0.12 µg/m^3^ decrease in PM_2.5_, and a 1% increase in “heavy industry” results in a 0.07 µg/m^3^ increase in PM_2.5_ concentrations. Although the overall results from these 25 sites are within the annual ranges established by the EPA, they reveal substantial variations that reinforce the value of hyper-local sensing technologies as a powerful surveillance tool.

## 1. Introduction

Particulate matter 2.5 (PM_2.5_), defined as particle mass with an aerodynamic diameter that is less than 2.5 µm, is regulated by the United States Clean Air Act and reported as micrograms per meter cubed (µg m^−3^), which is a mass-based concentration [1]. PM_2.5_ is typically found in higher concentrations in densely populated regions [2], and in combination with other air pollutants such as nitrogen dioxide (NO_2_) and O_3_ (Ozone), has been associated with serious health effects and an increased risk of mortality [3,4]. PM_2.5_ was ranked #1 as a risk factor for deaths in China and #6 as a risk factor for deaths in the US [3]. Using an atmospheric chemical transport model, Anenberg et al. (2010) estimated that global anthropogenic PM_2.5_ was associated with millions of cases of cardiopulmonary mortality and thousands of cases of lung cancer mortalities, but when reduced to a lower concentration of 5.8 µg m^−3^, the mortality estimates were reduced by approximately 30% [5]. Between 2010 and 2012, roadway air pollution alone in 42 New York City neighborhoods resulted in 320 deaths and 870 hospitalizations and emergency room visits due to PM_2.5_ [6]. The overlap of high particulate matter, high population, and poor health quality in the Midwestern region of the US results in a high incident of premature mortality [7].

Due to the serious human health impacts, the US Environmental Protection Agency (EPA) requires states to monitor exposure to PM_2.5_ and other gases. The EPA has set exposure limits of PM_2.5_ at 35 µg/m^3^ in a 24-h cycle that is averaged over three years, and a 12 µg/m^3^ annual standard limit. However, many studies indicate that chronic human exposure to levels of air pollution below the EPA standards results in a positive association with adverse health effects including shaping DNA methylation through epigenetic mechanisms, which can have multi-generational effects. Humans are most susceptible to such impacts during pregnancy, childhood, and the elderly stages of life [8,9,10,11,12,13]. Even though anthropogenic pollution from vehicles and industry result in increased PM_2.5_, there is also a correlation between PM_2.5_ and meteorological variables, which can explain up to 50% of the daily variations in some regions. For instance, on stagnant days, the average PM_2.5_ concentrations tend to be 2.6 µg/m^3^ higher [14,15]. Independent of geography, age, or gender, increasing long-term PM_2.5_ by 10 µg/m^3^ of PM_2.5_ increases non-accidental mortality by 6% [16]. A limitation of the studies that use PM_2.5_ in relation to human health outcomes is that due to the high cost and maintenance of regulatory monitors, most US cities have few monitors to gauge air quality and health burdens, sometimes just one or two sensors for millions of people. Enhanced spatial and temporal detail in monitoring is critical as a host of studies show the important role that local sources play in driving air pollution [14,17,18].

Scarcity of regulatory monitors has led to using satellite data to extend spatial coverage [19,20]. The deployment of low-cost in situ sensors in air quality studies has also increased with the development of more accurate monitoring devices, which has enhanced analysis of fine-scale variations [21,22,23]. Although low-cost devices have analytical drawbacks, they have been used effectively in research [24,25]. Low-cost PA sensors in particular perform well with respect to EPA regulatory monitors and have been used in several air quality studies [26] as a cheaper alternative to regulatory monitors to examine indoor and outdoor PM_2.5_ [24,27]. The interest in low-cost sensors grew as the need to assess and evaluate personal exposure to airborne pollutants and their impacts on humans and communities became clearer [1,28,29]. For example PA sensors have been found to have high self-consistency and can be used to fill in the gaps from sparse coverage available from regulatory grade sensors [30]. These sensors cost under USD 300, and have been used in cities such as Ft. Collins and Pittsburgh to examine air pollution as well as in a unique study to examine the association of the deployment and demographics in parts of California [30,31,32].

The goal of this study was to measure the spatio-temporal distribution and variability of outdoor air quality using 25 PA units that record and report real-time PM_2.5_, temperature, and humidity in Marion County, Indiana (Figure 1). Sensors were deployed from August 2018 through November 2019 and the resulting data were used to analyze the relationships between PM_2.5_ and various meteorological, land-use, and census variables in the Pleasant Run airshed in Indianapolis, Indiana. We examined: (1) the factors associated with daily averages of PM_2.5_ values that exceeded World Health Organization (WHO) guidelines of 25 µg/m^3^ (in 2021, this was changed to 15 µg/m^3^) [33], (2) locations with the highest odds ratio of exceeding the daily average of 25 µg/m^3^, and (3) the impact of tree canopy percentage on PM_2.5_ averages at the census tract level.

## 2. Materials and Methods

### 2.1. Sensor Network

We conducted this research in Indianapolis, partly because it has the characteristics of many large Midwest US cities (e.g., flat topography, multiple sources of PM), but also because it is local, allowing easier access for monitor deployment, QA/QC analysis, and the troubleshooting of sensors. Moreover, our research partner, Keep Indianapolis Beautiful (KIB) had access to citizen science networks from which we recruited volunteers to host the sensors. KIB is a local community-based nonprofit organization with a focus on improving the environment. We identified a list of individual households to contact and recruited 32 citizen scientists within the study area. We installed 32 PA-II-SD sensors, with Wi-Fi capabilities, over a roughly 96 (12 × 8) km^2^ study area in the eastern part of Indianapolis in Marion County, Indiana, to collect PM_2.5_ data (Figure 1). One of the sensors was installed on a three-story balcony. We are assuming that at such a height there may only be a small monotonic gradient difference between the ground and that level for PM_2.5_, based on previous research [34] and our own assessment of monitor performance in comparison to other monitors installed on the first floor of structures. All the sensors, as per the recommendations of the manufacturer, were installed under at least an overhang to provide some protection against the weather. The installation height of all but one sensor ranged from approximately 4 to 8 feet (1.2–2.4 m) above the ground, based on the availability of power outlets and overhead coverage at each location.

PA sensors are designed with a fan that draws a sample of air past its two independent laser counters labeled ‘Channel A’ and ‘Channel B’. Light from a particle is reflected to a detection plate where it is measured by a pulse. Particle size is determined by the length of each pulse, and particle count is determined by the number of pulses. Airborne particulate matter can include organic particles, inorganic particles, smoke, or dust. Particle sizes of 0.3, 0.5, 1, 2.5, 5, and 10 µm are counted and used to calculate mass concentrations (µg/m^3^) of PM_1.0_, PM_2.5_, and PM_10_ using an algorithm that was developed by Plantower for the PMS5003 sensor, a factory-calibrated instrument that estimates the number of suspended air particles based on the method outlined above. This compensates for the varying densities of various sources of PM_2.5_. PM_2.5_ readings are averaged and reported every 120 s on an interactive website from which they can also be downloaded. A chip in the sensors (ESP8266) provides the Wi-Fi capabilities enabling the uploading of its collected data to storage in the cloud provided by ThingSpeak. As needed, PA utilizes the sensor’s firmware to remotely check the devices and modify its software. An additional BME280 sensor is also incorporated into the PA unit that measures temperature, humidity, and pressure. The correction factor that the PA utilizes to adjust temperature and humidity before reporting it on their interactive map cannot be used universally. This technology, which uses a 5 v USB power source, offers a cheaper option than federal reference monitors to report PM_2.5_. Additionally, PA sensors also report relative humidity and temperature readings [35,36].

The PA sensors evaluated by South Coast AQMD against Federal Equivalent Method instruments costing over USD 20,000 found that laser b (Channel B) reported slightly higher PM_2.5_ mass concentrations; however, the two independent laser counters had a coefficient of determination of over 0.90 (R-square) tested over a range from 0 to 250 µg/m^3^, even though the maximum range is ≥1000 µg/m^3^ based on the technical documentation. The accuracy of the PA sensors with FEM between 0 and 250 µg/m^3^ was between 54.3 and 96.1% (moderate to good accuracy). The maximum consistency error for PM_2.5_ was found at ±10% at 100–500 µg/m^3^, and ±10 between 1 and 100 µg/m^3^ [36,37]. Due to a much higher number of rows of missing data (9197), we used the output from Channel B instead of Channel A in this study.

### 2.2. Processing PM_2.5_ Data

The PA sensors were deployed from August 2018 through November 2019, covering 20 census tracts on the east side of Indianapolis. The US Census Bureau (https://www.census.gov/ (accessed on 9 January 2023)) delineates census tracts, the geographical boundaries within a county of each state, with an ideal population of 4000 to gather and report statistical data (https://www.census.gov/programs-surveys/geography/about/glossary.html#par_textimage_13 (accessed on 9 January 2023)). Data from 25 of the 32 PA sensors (not in sequential order), deployed over an 11 month period from November 2018 to October 2019, was used in this study. A recall of power cords by the manufacturer in February 2019 resulted in approximately one lost data month as the sensors were disconnected and new cords were delivered and installed in all units. Hourly data from each sensor collected over the study period was downloaded from the PA website and we utilized Base-R in open-source software R-Studio version 4.0.3 (Posit, Boston, MA, USA), an open-source statistical computing software program, for data processing and analysis. Additionally, to validate the output from the PA, at deployment we compared raw PA online readings with an EPA-grade portable sensor (Thermo Scientific MIE pDR-1500).

PA units do not adjust for humidity, which impacts the hygroscopic growth of particles and particle count [30,38]. To reduce this humidity effect and account for the hygroscopic growth, the PM_2.5_ data were corrected using the formula in Equation (1) [32]:(1)$${\mathrm{C-PM}}_{2.5}=\frac{{\mathrm{PM}}_{2.5}}{1+\frac{0.25{\mathrm{RH}}^{2}}{\left(1-\mathrm{RH}\right)}}$$

In Equation (1), C-PM_2.5_ is the corrected PM_2.5_ that is reported by PA in µg/m^3^, PM_2.5_ is the raw PM_2.5_ value reported by PA (µg/m^3^), and RH is relative humidity retrieved from the Indiana Department of Environmental Management’s (IDEM) regulatory air quality monitor at Washington Park in Indianapolis. After this adjustment, the correlation between the data of the three closest sensors (Sensor 25, 30, and 31) to the two IDEM monitors (Washington Park monitor and I-70 monitor) was also examined. The Washington Park (WP) location is the only location that has a photochemical assessment monitoring station (PAMS), providing hourly samples of data that are available via the IDEM website from 1999 to present day [39]. Additionally, the I-70 monitor had more missing data than the WP monitor. Utilizing the WP monitor instead of the PA data in the above equation does introduce some uncertainty into this calibration process. However, since this is a very small region of the city close to the WP monitor, we believe that it is more appropriate to make this adjustment than either not calibrate the data or utilize PA relative humidity values, which are not accurate.

### 2.3. Meteorological Data

Some abnormal fluctuations in the temperature and humidity data were detected in the PA sensors in our study area. A study in Pittsburg (USA) concluded that due to the shape of the PA units, they can trap heat and increase the inside temperature by an average of 2.7 °C (36.86 °F) and lower the humidity by an average of 9.7% versus outside the unit [30]. This necessitated downloading meteorological data (temperature, relative humidity, windspeed) from the IDEM website reporting data from the air quality monitor at WP (https://www.in.gov/idem/airquality/ (accessed on 20 December 2020)). They have recently changed their reporting system, which now requires a request to the air monitoring department of IDEM for data needed for analysis. WP data are continuously monitored by a photochemical assessment monitoring station and electronically transmitted to IDEM for processing. Meteorological factors such as relative humidity, windspeed, and drought conditions play a role in PM_2.5_ concentrations [14,40,41,42].

### 2.4. Land Use and Land Cover

The estimates of tree canopy cover (TC) in the study region were based on a 1 m resolution land cover raster produced by our partner KIB and the University of Vermont Spatial Analysis Laboratory. The land cover data were derived from the classification of National Agricultural Imagery Program (NAIP) data acquired in 2013 and LiDAR data acquired in 2009. This dataset was used to quantify the tree canopy cover in the study region census tracts and within buffers at varying distances from the PA sensor locations.

Additionally, the distribution of “Heavy Industrial” land use was modeled using a vector-based GIS dataset from the City of Indianapolis’s data portal [43] and processed using the mapping software Maptitude 2020 by Caliper^®^. This polygon layer, with land use information, is utilized by the city of Indianapolis for zoning and planning. This particular variable was incorporated into the analysis to capture its impact on this study area [44].

### 2.5. American Community Survey

The most recent American Community Survey (ACS) 5-year estimates (2014–2018), at the census tract level, were used for socioeconomic and demographic variables [45]. Census tract data have been used in the past in conjunction with PA to examine environmental justice implications [46].

Statistical analysis including correlation, logistic regressions, and stepwise regression incorporating all of the variables discussed above (Appendix A and Appendix B) was conducted using the open-source software RStudio version 4.0.3. The binary response variables for logistic regression represent the success of days at sensor locations that exceeded the WHO daily limit (1) versus days that did not exceed the limit (0). The probability of an outcome varies based on the values of the explanatory variables.

## 3. Results

Sparse networks of regulatory monitors established by states are inadequate in providing us with local information, limiting the use of these data to make informed decisions on zoning and the infrastructure required to build robust communities. A dense network of sensors, as established in this study, enables us to examine local impacts that can then be utilized to make future informed decisions. The temporal array for each sensor spans thousands of data points, but the spatial array is limited. Due to a limited dataset, we did not examine a spatial relationship among the sensor arrays.

One clear example of local monitoring to assess event-based air quality variations involves smoke emissions from US Independence Day celebrations on 4 July 2021. As a result, sensor 4, as shown in Figure A1, and other sensors revealed higher daily PM_2.5_ averages due to local fireworks than the values reported later that month (second highest peak) that resulted from massive forest fires in the Pacific Northwest and southern Canada [47,48,49].

### 3.1. PA Data and Portable EPA Grade Sensor

At deployment, PA validation with the portable EPA grade sensor indicated that, on average, the values for 21 of the 25 sensors deployed were 13.56% higher than the EPA grade sensor, which is within a relative accuracy of +/−20% [30]. It is important to note that the PA sensors’ factory calibration is based on specific ambient aerosol, which may not be identical to our study region and could have contributed to the observed differences. The calibration of raw data has been shown to reduce errors by as much as 25% for extreme cases and by 10% for typical cases. Additionally, there is a systematic bias between instruments that can be compounded with the variation of particle composition and the sensors’ performance in the field, which can be balanced by longer term (one year) averaging, thus significantly reducing the error [30]. Our field validation was limited with the EPA grade portable sensor, typically lasting just ~1 h at the beginning of deployment.

### 3.2. Correlation of PA Data against IDEM Monitor Data

Spearman’s correlation was used to examine the relationship between the PM_2.5_ measurements collected at the two IDEM monitoring sites and the adjusted PM_2.5_ data collected at the three closest PA sensors (labeled 25, 30, and 31; Figure 2). This calculation was based on data spanning 11 months with over 300 observations each for the three sensors (Table 1).

The association between PM_2.5_ readings at the three PA sensors was strongest with the IDEM Washington Park sensor. The range of raw PM_2.5_ was 0.36–103.80 µg m^−3^ with RH ranging from 44 to 99%, and the average temperature was between −2.7 and 85.2 °F (−19 to 30 °C), based on the WP sensor data. After applying the relative humidity correction, this range adjusted to 0.08–80.14 µg m^−3^, the mean values to 16.22–9.03 µg m^−3^, and the median to 14.64–8.10 µg m^−3^. This RH correction underestimation may be more pronounced due to the higher humidity in Indianapolis most of the year versus a less humid environment such as that examined in Colorado, where the median bias was reduced by −15% after the correction [32].

Table 2 shows that when the WP monitor’s relative humidity values are used in Equation (1) to correct PM_2.5_, the WP-adjusted values are less than 2% different from the WP data. This indicates that, based on the chemical composition of the pollutants in this region, an RH-based correction (Equation (1)) is adequate for calibration, which is what was used for all further analyses in this study. The mean value of PM_2.5_ is lower in this network of 25 locations than is reported in a prior study in Indianapolis [50]. One possible explanation could be that the sensors in this study are predominantly in residential areas versus being near or on major thoroughfares.

### 3.3. Driving Factors Impacting Daily Averages of PM_2.5_ Exceeding WHO Limits

A total of 8124 counts of cumulative data by month were used in the regression analysis (Table 3). We recognize that the grouping of the months is subjective, but we grouped them, as outlined in Table 3, to best capture the seasonal variations in the study. To identify the potential factors impacting the days that PM_2.5_ values exceed the WHO threshold of 25 µg/m^3^, the independent variables listed in Table 4 (Appendix A and Appendix B) were used in setting up a logistic regression equation. The dependent binary variable is represented by 1 when the adjusted PM_2.5_ values are greater than or equal to the WHO limit of 25 µg/m^3^, and 0 when the adjusted PM_2.5_ values are less than the WHO limit of 25 µg/m^3^.

Logistic regression output and its significance based on *p*-values is represented in Table 4 for the days that exceed the WHO limit versus the days that do not. This model indicates that five variables (precipitation, windspeed, Month group 5, Tuesday, and Saturday) are significant in impacting the days exceeding the daily WHO limit (Table 4). Month groups 2–5 are in reference to Month group 1, indicating that all else being equal, Month group 5 is significant when compared to Month group 1. Saturday through Thursday are referenced against Friday. For ease of interpretation, the significant variables from Table 4 are further extrapolated in Table 5 into the odds of a day exceeding the daily WHO limit. Any odds under 1 are represented by reduced odds and over 1 by increased odds of resulting in a high PM_2.5_ concentration exceeding the WHO daily guidelines. Increased precipitation, increased windspeed, and Month group 5 (compared to Month group 1) have reduced odds of exceeding the WHO guidelines per day (Table 5). On Saturday, the odds of exceeding the WHO limit are sixfold compared to the reference day (Friday), and on Tuesday those odds decrease to threefold.

### 3.4. Identify Sensors with Highest Odds Ratio of Exceeding the WHO Daily Limit

Odds ratio calculations were run to identify locations within the study area that are the likeliest to exceed the WHO daily limit for PM_2.5_. There were only two sensors out of a total of 25 that had significant odds ratios for days that exceed the WHO daily limit of 25 µg/m^3^. The odds for sensor 16 are 3.04 times that of other locations to meet or exceed 25 µg/m^3^, and for sensor 13 it is 2.37 (Table 6). Interestingly, the two sites proximal to major interstate freeways in the area (sensors 32 and 25) had low odds ratios for exceeding the air quality parameters, perhaps owing to freeway turbulence, lack of idling, and modern emission controls for most vehicles.

### 3.5. Analysis at the Census Tract Level

Correlations at the census tract level aggregation of PM_2.5_ data, census data, tree canopy data, and GIS data revealed that four variables are significant (*p*-value 0.05), with one being almost significant at that level (Table 7, Appendix B). The two negative relationships with Tree Canopy % and White Only % indicate that they have an inverse relationship with PM_2.5_ concentrations. The two positive correlations were found with ‘Heavy Ind %’ and ‘Hwy Length km’ with PM_2.5_ concentrations.

Regression models were run with all of the variables (Table 8, Appendix B). Model 1 resulted in an adjusted r-square of 66.47% with four significant variables at a *p*-value < 0.05. However, this model resulted in a high variance inflation factor (VIF) of 176.17 for ‘Black One Race %’, which led to running Model 2 without this variable. The second model with an adjusted r-square of 57.94% resulted in a high VIF value of 12.57 for ‘Road Length km’. Model 3 was run without ‘Road Length km’, which resulted in an adjusted r-square of 59.92%, higher that Model 2, but lower than Model 1, with all VIF values < 10. The output of Model 3 is detailed in Table 8, which shows four significant variables at *p*-value < 0.05, and population with some college and associates degree at *p*-value 0.1. As an additional step, stepwise regression model was run on Model 3, which resulted in an adjusted r-square of 62.79% with two significant variables at *p*-value < 0.01 and three variables significant at *p*-value < 0.05. All VIF values for the variables in the final model (stepwise model) are under 10, indicating that multicollinearity is not an issue in this model. We can make the following observations from the final model for our data points at the census tract level:A 1% increase in canopy coverage deceases average PM_2.5_ at the census tract level by 0.12 ± 0.03 µg/m^3^ (at the 95% confidence interval).A 1% increase in Heavy Industrial area increases PM_2.5_ by 0.07 µg/m^3^ ± 0.02 µg/m^3^.As the percentage of population over 25 with some college and associates degree increases, it results in a proportional increase in PM_2.5_ of 0.08 µg/m^3^ ± 0.03.Hispanic Latino % has a proportional increase, indicating an increase in this population by one percent results in an increase in PM_2.5_ of 0.06 µg/m^3^ ± 0.02.Median Rent has an inverse relationship. An increase of USD 100 in median rent results in a decrease in PM_2.5_ of 0.9 µg/m^3^ ± 0.03.

## 4. Discussion

Community engagement in meteorology may have its earliest examples dating back to the 1840s [51]. At any level, citizen science poses many challenges as well as opportunities [52]. By creating a link between researchers and the public, such engagement not only benefits the researchers, but also creates a more engaged citizenry [51]. Low-cost sensors such as PA are an inexpensive way for communities to use citizen science to participate in examining air quality at a fine scale [21].

The South Coast AQMD has tested PA sensors to evaluate their performance and found high field accuracy. Our findings generally corroborate this instrumental fidelity, with the correlation between the three sensors tested against the Washington Park sensor of >0.7. Over time, sensor fidelity does degrade [53], along with the added confounding effects of temperature and humidity. Thus, low-cost monitoring data need to be carefully examined over time to ensure that the calibration equation is adequately addressing any data deviations. To suppress the sensor from humidity [24], we calibrated the PA sensor output with humidity values (Equation (1)) from the Washington Park IDEM monitor [54]. This is a limitation of this study, since we were unable to utilize humidity data from each sensor location; however, we believe that, given the study area, utilizing accurate humidity data from the WP monitor was adequate. We also found that a relative humidity correction alone in the PR dataset was adequate (Table 2). We selected several census tracts for this comparison, as this analysis was used as an illustrative example of how AQ related to public health costs rather than a comprehensive analysis of this relationship, which is beyond the capabilities of this sensor array and our experimental approach.

In our study area, Saturday and Sunday were in the top three days for exceeding WHO PM_2.5_ standards. This may be due to a higher volume of vehicle trips during these days on local roadways, with an increase in idling time as well as stop-and-go traffic. More vehicle traffic on residential roads proximal to the sensors during weekends would be captured by the sensors, as opposed to standard work commute days when the traffic would be more limited to traveling along major arterial roads to and from work. A previous study in Indianapolis (Sullivan & Pryor, 2014) found higher PM_2.5_ during weekdays, in contrast to our results, but this is likely driven by the fact that this study used stationary sites from IDEM monitoring sites that are intentionally placed distant from local air pollution sources whereas the previous study sampled air quality using transects that were proximal to freeways and major arterials as opposed to neighborhood-based sites.

We expected that sensors placed near major highways (sensors at location 32 and 25), would have high odds of exceeding WHO limits, but we found instead that sensors 13 and 16 had the highest odds of exceeding daily limits. We expect that the exhaust from traffic on highways creates a temperature gradient between it and the ambient air, which, due to thermal buoyancy causes the plume to rise, which is then impacted by windspeed and direction. Lower windspeeds versus high windspeed on highways result in traffic exhaust plumes dispersing more slowly, thus resulting in higher measurements that are detected for longer periods of time [55]. However, since sensor 32 is upwind from I-65, it is impacted by local versus highway traffic. Sensor 16, on the other hand, along with vehicles, is also impacted by the frequent use of a wood burning stove by its neighbor, thus having a unique local impact, resulting in consistently higher values at that sensor location.

Urban greening initiatives in census tracts with low tree canopy cover can positively impact air quality and reduce related health disparities. Tree canopy coverage will become increasingly important, as urban land is projected to increase to 8.1% in 2050 versus 3.1% in 2000. Indiana’s urban land is projected to increase from 8.8% in 2000 to 16.7% by 2050. In 1990, 2.6% of Indiana’s tree canopy coverage was in urban areas. This percentage increased to 3.6% in 2000 and is expected to increase to 12.1% by 2050. By 2050, 8.8% of Indiana’s tree canopy coverage outside of urban areas will be subsumed by urban growth. This projection equates to 1500 km^2^ of land being subsumed by urbanization between 2000 to 2050. Since forests and trees are critical in enhancing human and environmental health, urban canopy cover should be prioritized [56]. Not only does reduced PM_2.5_ improve air quality, but it also reduces expenses. A 10-city study in the USA found that mortality related to PM_2.5_ ranged between 1 and 7.6 people/year, and the average value per mortality incidence was USD 7.8 million. Additionally, the average health benefits were USD 1600 per hectare of tree cover, with an average of USD 1.6 billion in health benefits per 1µg/m^3^ reduction [57]. According to this study, the health savings in our 20 census tracts ranged from approximately USD 36,300 (census tract 18097355900) to USD 1,121,585 (census tract 18097361400). This was calculated by multiplying the canopy coverage in each census tract by cost savings of USD 1600 as reported by Nowak et al. (2013). When this calculation is normalized by area, the health savings in census tract 18097361400 is 1.2 times that of the former.

Planting trees in the census tract with lower canopy coverage in conjunction with dividing the study area into low-emission zones (LEZs), where there are restrictions placed on high-polluting vehicles from entering, could give us direct control over minimizing the impacts from vehicle pollution [58] and combatting it through nature simultaneously.

## 5. Conclusions

This air quality study in Indianapolis is a community-based attempt to examine local impacts of PM_2.5_ over a dense network of 25 sensors. This network was established by collaborating with local partners through community engagement and enhancing the examination of the impacts of canopy coverage along with land use and other variables on PM_2.5_ concentrations in the city.

We found that the relative humidity correction, as captured in Equation (1), is adequate for calibrating raw PM_2.5_ data that is used in all the analysis. Increased percentage of tree canopy coverage at the census tract level was associated with lower PM_2.5_ concentrations. A 1% increase in canopy coverage at the census tract level resulted in lowering PM_2.5_ by approximately 0.12 µg/m^3^. Based on research by Nowak et al. (2013), we extrapolate further that the canopy coverage in our study region provides between USD 36,000 and USD 1,121,585 in health savings. Additionally, a 1% increase in the Heavy Industrial area classification in the census tract resulted in increasing PM_2.5_ by a modest 0.07 µg/m^3^.

In our logistic regression analysis, we found that increased windspeed and precipitation was associated with lower PM_2.5_ concentrations. We also found that local impacts, as witnessed by the wood burning stove near sensor 16could exceed concentrations measured at locations near major highways. Lastly, because our sensors were deployed in predominantly residential areas, our finding that Saturdays had the highest accumulations of these particles may be due to increased neighborhood traffic on the weekends. Wind direction, which impacts pollutant concentrations, was not modeled in this paper, but should be included in future research.

Air pollutants such as PM_2.5_ can vary both spatially and temporally based on the source(s), atmospheric conditions, and built environment features [59]. Integrating data from dense sensor networks like the one established in our study along with health outcomes data can facilitate understanding of spatial variability in air quality. It can also serve as a tool to facilitate partnership between the community and government entities to quantify and improve air quality as well as inform policies that contribute to creating more resilient communities.

## Figures and Tables

**Figure 1 ijerph-20-01934-f001:**
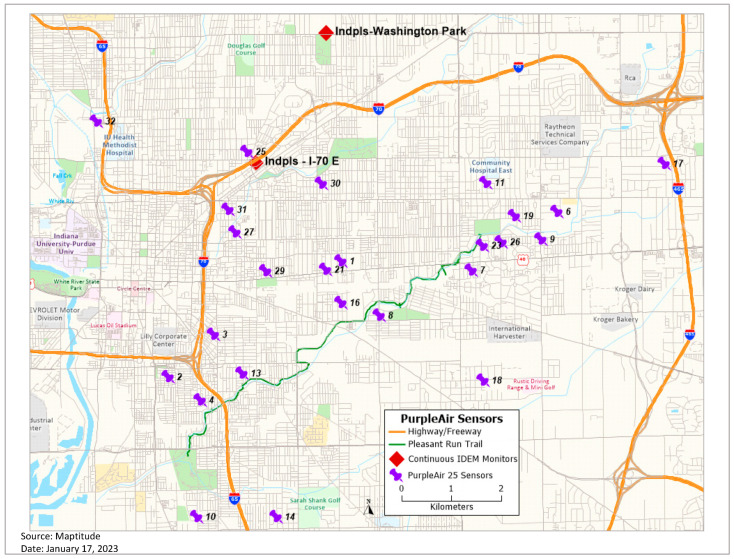
Locations of 25 PA sensors in the study area.

**Figure 2 ijerph-20-01934-f002:**
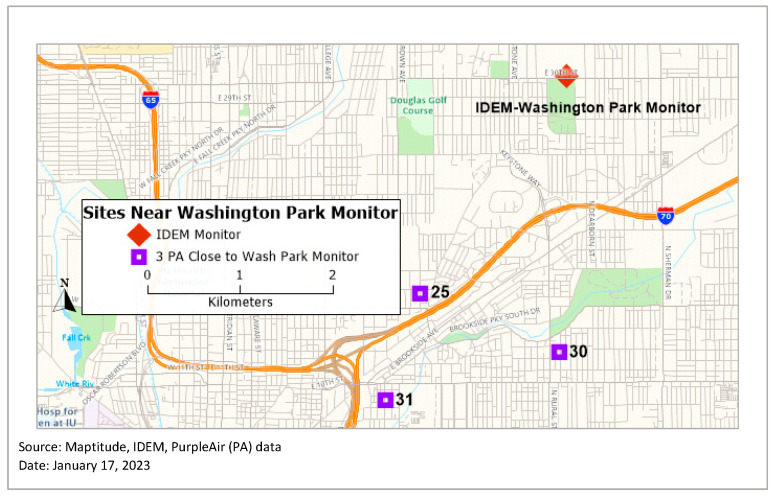
Sensors nearest to Indiana Department of Environmental Management (IDEM) regulatory monitors.

**Table 1 ijerph-20-01934-t001:** Correlation of sensors 25, 30, and 31 with IDEM monitors and PM_2.5_ data.

Sensor Number	IDEM Site	Correlation (Spearman)	*p*-Value
25	WP	0.75	<2.2 × 10^−16^
30	WP	0.74	<2.2 × 10^−16^
31	WP	0.72	<2.2 × 10^−16^
25	I-70	0.72	<2.2 × 10^−16^
30	I-70	0.71	<2.2 × 10^−16^
31	I-70	0.72	<2.2 × 10^−16^

**Table 2 ijerph-20-01934-t002:** Results from applying Equation (1) to PA data.

Variable	Median (µg m^−3^)	% Diff from WP PM_2.5_	Unit Diff from WP PM_2.5_
Raw PM_2.5_	12.874	36.45%	3.439
WP adjusted PM_2.5_ (Equation (1))	9.616	1.92%	0.181
WP PM_2.5_	9.435	0.00%	0.000

**Table 3 ijerph-20-01934-t003:** Logistic regression observation breakdown by month for 25 deployed sensors.

Month	Month Group	Observations
2018		
November	1	728
December	2	769
2019		
January	2	739
March	3	753
April	3	726
May	3	773
June	4	702
July	4	752
August	4	751
September	5	733
October	5	698
Total		8124

**Table 4 ijerph-20-01934-t004:** Logistic regression output. Negative values indicate a decrease, and positive values an increase, in the number of days with PM_2.5_ averages above WHO guidelines.

Variable	Estimate	Pr (>|z|)
(Intercept)	−2.05301	0.0014 **
Precip cm	−34.137	0.0060 **
Windspeed kmh	−0.27462	0.0000 **
Temp C	−0.02154	0.2569
Month group 2	−0.58128	0.1487
Month group 3	0.10966	0.7550
Month group 4	−1.08057	0.0518
Month group 5	−1.1233	0.0214 *
Day (Monday)	0.5873	0.2822
Day (Saturday)	1.84268	0.0002 **
Day (Sunday)	1.01762	0.0509 .
Day (Thursday)	0.09561	0.8816
Day (Tuesday)	1.17912	0.0184 *
Day (Wednesday)	−1.07063	0.2030

“.” signifies *p*-value < 0.1, “*” signifies *p*-value < 0.05, “**” signifies *p*-value < 0.01.

**Table 5 ijerph-20-01934-t005:** Odds of the significant coefficients from the regression output.

Variable	Odds
Precip cm	0.0000
Windspeed_kmh	0.7599
Month group 5	0.3252
Day (Saturday)	6.3134
Day (Sunday)	2.7666
Day (Tuesday)	3.2515

**Table 6 ijerph-20-01934-t006:** Odds output from significant sensors.

Sensor#	chi_sq (*p*-val)	Significant Status	OR
13	0.0243	Signif Relationship	2.37
16	0.0008	Signif Relationship	3.04

**Table 7 ijerph-20-01934-t007:** Correlations of average PM_2.5_ at the census tract level.

Correlations with PM_2.5_ Census Tract Level	Method	Value
Tree Canopy %	Spearman	−0.67 *
Heavy Ind %	Spearman	0.63 *
Hwy Length km	Spearman	0.50 *
Road Length km	Spearman	0.29
Pop25 + LT high school %	Spearman	0.44 ^a^
Pop25 + Some College_Assoc %	Pearson	−0.10
Pop25+ Graduate_Prof Degree %	Spearman	−0.20
Hispanic Latino %	Spearman	0.24
Black One Race %	Spearman	0.30
White Only %	Spearman	−0.47 *
Median HH Inc.	Spearman	−0.41
Median Rent	Pearson	0.10

‘*’ represents *p*-value < 0.05, ‘^a^’ represents almost significant at *p*-value 0.05.

**Table 8 ijerph-20-01934-t008:** Stepwise regression output.

	Model 1 (Adj-R^2^ = 66.47)	Model 3 (Adj-R^2^= 59.92)	Stepwise (Adj-R^2^ = 62.79)
(Intercept)	32.1600 *	16.0000 **	16.66913 **
Tree Canopy %	−0.1840 *	−0.1165 *	−0.1243 **
Heavy Ind %	0.0796 *	0.0724 *	0.069337 *
Hwy Length km	−0.1284	0.0225	
Road Length km	0.0097		
Pop25 + LT high school %	−0.4331	−0.4173	−0.48426
Pop25 + Some College_Assoc %	0.1603 *	0.0790	0.080368 *
Pop25 + Grad Prof Degree %	−0.0131	0.0634	0.044515
Hispanic Latino %	−0.0831	0.0590 *	0.060178 *
Black One Race %	−0.1201		
White Only %	−0.0860	0.0167	0.01149
Median HH Inc	0.0000	0.0000	
Median Rent	−0.0153 *	−0.0083 *	−0.00873 **

“*” signifies *p*-value < 0.05, “**” signifies *p*-value < 0.01.

## Data Availability

The data presented in this study are openly available at ScholarWorks at IUPUI (https://scholarworks.iupui.edu/handle/1805/30643 (accessed on 1 December 2022)).

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
