# Peer review of "Efficacy of Low-Cost Sensor Networks at Detecting Fine-Scale Variations in Particulate Matter in Urban Environments"

_ijerph, 2023, doi:10.3390/ijerph20031934_

Round 1

Reviewer 1 Report

In general, this paper is innovative in carrying out regional-scale PM2.5 research based on low-cost sensors. However, the accuracy of the research data needs to be improved to support the study of the spatial pattern of PM2.5. Most of the research methods are mathematical statistics, which do not fully combine the data with the spatial position. And the conclusions are not sufficient to guide the optimization of the air quality spatial pattern for PM2.5. It is suggested to optimize the research process and carry out further researches.

1. Chapter 1 (line 73-76) lacks the comparison between PA sensors and EPA regulatory monitors in terms of cost and maintenance, and does not reflect the advantages and necessity of PA sensors for measuring PM2.5 data. Further elaboration is suggested.

2. Chapter 1 (line 82-83) mentions that the PA sensors have been used in some researches, but there is no more detailed introduction to these researches, and it is suggested to supplement.

3. Chapter 1 (line 91-93) points out that the goal of this paper is to measure and model variability of outdoor air quality. However, the paper does not carry out the above-mentioned related research, but studies the data characteristics and distribution characteristics of PM2.5 on a regional scale based on PA sensors’ data.

4. Chapter 2, the sensor network (line 112), mentions that one of the sensors is installed in a 3-story balcony. However, the influence caused by the difference in height should not be directly assumed, but should be fully discussed. Besides, the paper below does not explain and analyze the data of this sensor separately.

5. Chapter 3 (line 186-194) introduces that the spatial pattern of PM2.5 data cannot be obtained by kriging and Inverse Distance Weighting because the statistically valid sample size is not approach, indicating that the current research data cannot support the follow-up research in terms of correlation analysis. It is suggested to increase the effective sample size to improve the accuracy of the research.

6. Chapter 3 introduces that a large number of statistical methods including Pearson, Spearman, and logistic regression are used to study the relationship between PM2.5 and various factors. However, an introduction to the above statistical methods is missing in chapter 2, the materials and methods (line 101). It is suggested to add.

7. Chapter 3, the analysis at the census tract level (line 291-325), lacks a description of the spatial correspondence between the census data and PM2.5 data, including data coverage, data spatial precision, etc. It is suggested to add.

8. Chapter 3 lacks of explanation of the reasons why the factors including tree canopy, heavy industry, road length, education level and race are chosen. It is suggested to add.

9. Chapter 3 (line 237-248) mentions using the equation 2 and 3 to correct the sensors’ data based on the methodology outlined by Malings et al. However, this part is not the conclusion of the paper and should be placed in chapter 2, the materials and methods.

10. Chapter 3 only conducts a simple analysis of sensor 13 and 16 data from the perspective of odds ratio (lines 280-290), and there are few studies on the spatial distribution characteristics of sensor data. Further elaboration is suggested.

11. Figure 1, Figure 2, and Figure 3 in the main text of the paper have low resolution, especially the legend parts are blurry, and it is suggested to replace them with clearer ones.

12. In Chapter 5 (line 408-413), the part about the reasons for the abnormal measurement data of some sensors is not comprehensive compared with what has mentioned previously. It should also include the influence of wind direction on the data, and it is suggested to add.

Reviewer 2 Report

In substance, I can accept this paper. But there are a few things to confirm:

1. Why did you choose the study location there? What objective academic reasons confirm the need for a study at that location?

2. How to make use of research results? How do related parties utilize and use it?

3. And your method can be used in various places. However, the problem lies in the utility for local area managers.

4. What factors or variables have the most dominant influence on environmental conditions in your urban area?

Reviewer 3 Report

The manuscript has an issue of methodological, data, analytical transparency. Please improve those transparencies. Specific comments can be found in the attached file. 

Reviewer 4 Report

Comments on Heintzelman et al.

Heintzelman et al. used low-cost sensors to study the fine-scale PM2.5 variations in Indianapolis, Indiana, US. Major revisions are needed. Below are my comments:

1.      It seems RH data used for the calibration were from only one site. The uncertainties caused by neglecting the spatial differences in RH should be discussed.

2.      I don’t see the necessity of using Eq. 2 or Eq. 3 for the calibration.

3.      For the logistic regression analysis, why are Wednesday and Friday missing? A better explanation for Table 4 and Table 5 should be provided. For example, on a rainy Saturday, what are the odds of PM2.5 > 25 µg/m3?

4.      The seasonal/month differences in the canopy should be considered.

Some minor comments:

L39: Add “, and” after O3 (Ozone)

L41-43: The reference is missing.

L75: The phrase “a handful of” is informal, it should be rephrased.

Fig. 1: A larger map showing the location of Indiana with respective to other states should be added.

Fig. 2, which is hard to read, should be improved.

L193: If “neither approaches the sample size to be considered statistically valid”, what’s the point of performing the analysis?

Table 1: The meaning of the “IDEM site” should be explained. Also, using two significant numbers for the correlation are sufficient.

L255-256: Possible reasons should be given.

L267: “Table 4.5” should be “Table 5”.

Table 4: Why is “Tuesday” also significant?

Many terms (such as “Precip cm”, Windspeed_kmh”, “chi_sqin”, “Hwy Length km”, “Pop25…”) in the tables should be clearly defined.

Table 7: Why were different methods for calculation correlations used?

Table 8: Results from Model 2 are missing.

Reviewer 5 Report

This work investigated how particulate matter vary at the hyper-local scale, and the role that proximal sources play in influencing neighborhood-scale patterns. The authors examined PM2.5 variations in one airshed within Indianapolis (Indiana, USA). They analyzed PM2.5 with meteorological data, tree canopy coverage, land use, and various census variables. Results indicated that 1% increase in tree canopy at the census tract level results in a ~ 0.12 μg/m3 decrease in PM2.5, and a 1% increase in “heavy industry” results in a 0.07 μg/m3 increase of PM2.5 concentrations. 

1. What are the models used in R-Studio for analyzing the data? Please provide related references or equations in methodology section.

2. In the paragraph ling 186-194, why the two approaches are failed to generate valid spatial patterns in PM2.5? Are those data from sparse networks of regulatory monitors or the dense network of sensors?

3. What are the two regression models used for analyzing the variables in Table 8? Please provide explanation in Methodology section. 

4. Both PA and PurpleAir are existing throughout the whole article, which is a little bit confusing. Please replace all the PurpleAir, except the first one, to PA for conciseness. 

Round 2

Reviewer 1 Report

In general, the content of the paper is more complete after the revision, and there are still several key points that need to be revised, because this is crucial to the logical integrity of the paper. Specific comments are blow.

1. The application case of PA sensor lacks the relevant information such as the detection coverage, coverage density, and arrangement number of PA sensors, and how the related applications provide support and inspiration for the experimental protocol of this study, and it is recommended to supplement.

2. After the revision, the description of the spatial correspondence between the census data and the sensor data is still lacking. The correspondence is important and is the basis for conducting correlation analysis.

3. The author should explain why factors such as canopy rate, road mileage, education level and ethnicity should be selected for correlation analysis with PM2.5 data. This is scientific support for conducting correlation analysis.

4. Chapter 3 introduces that the spatial pattern of PM2.5 data cannot be obtained by kriging and Inverse Distance Weighting because the statistically valid sample size is not approach, indicating that the sensor data obtained so far is not enough to simulate the distribution of PM2.5 in the entire urban area. So the author needs to clarify the coverage that the current sensor can detect, and mark it on the figure.

5. The author should describe the spatial accuracy and coverage of data such as tree canopy, heavy industry, road length, education level and race in more detail in Chapter 2.

Author Response

Please see the attached file below.

Author Response

Please see the attached file below.

Reviewer 4 Report

The revised manuscript has improved significantly. I believe it can be accepted for publication after language editing.

Author Response

Thank you for your review.

Reviewer 5 Report

All my previous comments and concerns have been well addressed by the authors. Therefore, a publication of this article on IJERPH is recommended.

Author Response

Thank you for your review.